# IGFBP7 Drives Resistance to Epidermal Growth Factor Receptor Tyrosine Kinase Inhibition in Lung Cancer

**DOI:** 10.3390/cancers11010036

**Published:** 2019-01-02

**Authors:** Shang-Gin Wu, Tzu-Hua Chang, Meng-Feng Tsai, Yi-Nan Liu, Chia-Lang Hsu, Yih-Leong Chang, Chong-Jen Yu, Jin-Yuan Shih

**Affiliations:** 1Department of Internal Medicine, National Taiwan University Hospital, National Taiwan University, Taipei 10002, Taiwan; b8501091@gmail.com (S.-G.W.); thchang15@gmail.com (T.-H.C.); benson1032@gmail.com (Y.-N.L.); jefferycjyu@ntu.edu.tw (C.-J.Y.); 2Department of Internal Medicine, National Taiwan University Cancer Center, National Taiwan University, Taipei 10672, Taiwan; 3Department of Molecular Biotechnology, Da-Yeh University, Changhua 51591, Taiwan; tsaimf@mail.dyu.edu.tw; 4Department of Medical Research, National Taiwan University Hospital, National Taiwan University, Taipei 10002, Taiwan; chialanghsu@ntuh.gov.tw; 5Department of Pathology, National Taiwan University Hospital, National Taiwan University, Taipei 10002, Taiwan; ntuhylc@gmail.com; 6Graduate Institute of Clinical Medicine, College of Medicine, National Taiwan University, Taipei 10002, Taiwan

**Keywords:** *EGFR* mutation, insulin-like growth factor binding protein 7, epidermal growth factor receptor-tyrosine kinase inhibitor, acquired resistance, lung cancer

## Abstract

Patients with epidermal growth factor receptor (*EGFR*) mutation-positive lung cancer show a dramatic response to EGFR-tyrosine kinase inhibitors (TKIs). However, acquired drug resistance eventually develops. This study explored the novel mechanisms related to TKI resistance. To identify the genes associated with TKI resistance, an integrative approach was used to analyze public datasets. Molecular manipulations were performed to investigate the roles of insulin-like growth factor binding protein 7 (*IGFBP7*) in lung adenocarcinoma. Clinical specimens were collected to validate the impact of IGFBP7 on the efficacy of EGFR TKI treatment. *IGFBP7* mRNA expression in cancer cells isolated from malignant pleural effusions after acquired resistance to EGFR-TKI was significantly higher than in cancer cells from treatment-naïve effusions. IGFBP7 expression was markedly increased in cells with long-term TKI-induced resistance compared to in TKI-sensitive parental cells. Reduced IGFBP7 in TKI-resistant cells reversed the resistance to EGFR-TKIs and increased EGFR-TKI-induced apoptosis by up-regulating B-cell lymphoma 2 interacting mediator of cell death (BIM) and activating caspases. Suppression of IGFBP7 attenuated the phosphorylation of insulin-like growth factor 1 receptor (IGF-IR) and downstream protein kinase B (AKT) in TKI-resistant cells. Clinically, higher serum IGFBP7 levels and tumors with positive IGFBP7-immunohistochemical staining were associated with poor TKI-treatment outcomes. *IGFBP7* confers resistance to EGFR-TKIs and is a potential therapeutic target for treating EGFR-TKI-resistant cancers.

## 1. Introduction

Lung cancer is a leading cause of cancer death worldwide [1]. Constitutive activation of the epidermal growth factor receptor (EGFR) signaling pathway has been implicated in the initiation and progression of lung cancer. EGFR-tyrosine kinase inhibitors (TKIs) have recently been developed as therapeutic agents for patients with lung cancer carrying *EGFR*-activating mutations. The two most common *EGFR*-activating mutations are exon 19 deletions (del E746-A750) and a single substitution mutation in exon 21 at codon 858 (L858R) [2]. Approximately 10–20% of patients with non-small-cell lung carcinoma (NSCLC) in Western countries and 40–50% in East Asia have tumors harboring somatic activating mutations in *EGFR* [2,3].

Despite the demonstrated benefits of *EGFR*-targeting therapies, these tumors invariably develop drug resistance [4,5,6]. Various mechanisms are involved in the development of EGFR-TKI resistance, including secondary *EGFR* mutations, aberrance in downstream pathways, and activation of alternative pathways [7]. A secondary mutation in *EGFR* (T790M in exon 20), reported in 2005, enhanced the affinity of the ATP binding pocket for ATP, and thus conferred acquired resistance to EGFR-TKIs [8]. The *EGFR* T790M mutation is the most frequent cause of TKI resistance and is detected in approximately half of NSCLC patients after developing acquired resistance [5,8]. Additionally, amplification of the proto-oncogene *MET* (encoding the hepatocyte growth factor receptor) also contributes to EGFR-TKI resistance and is detected in approximately 20% of these patients [9,10]. Previous studies indicated that the epithelial–mesenchymal transition regulator, Slug, confers resistance to EGFR-TKI therapy in lung adenocarcinoma patients and that interleukin (IL)-8 plays a role in the EGFR-TKI-resistance mechanism by regulating cancer stem cell properties [11,12]. Although some resistance mechanisms have been identified, additional information is needed to understand and overcome the resistance to EGFR-TKI therapies.

To facilitate the development of effective therapies against NSCLC, we explored the novel mechanisms of EGFR-TKI-resistance underlying tumor progression. An integrative approach was used to analyze public datasets. The results revealed that insulin-like growth factor binding protein 7 (*IGFBP7*) expression affects TKI resistance. Molecular manipulations were conducted to investigate the roles of *IGFBP7* in TKI-resistant lung adenocarcinoma. Additionally, clinical specimens were collected to validate the impact of IGFBP7 on the efficacy of EGFR-TKI treatment. Our findings indicate that *IGFBP7* plays a role in the mechanisms of resistance to EGFR-TKIs and is a potential target for overcoming EGFR-TKI resistance.

## 2. Results

### 2.1. IGFBP7 Was the Highest-Ranking Gene Related to TKI Resistance in Public Datasets

To identify the genes associated with TKI resistance, we collected four gene expression datasets consisting of 15 experiments to compare TKI-sensitive to TKI-resistant cells, and calculated the TKI resistance-related score for each gene based on its expression (Appendix A) [13,14,15]. In each dataset, corresponding TKI-resistant cell lines were generated from TKI-sensitive cell lines (PC9, HCC827, and HCC4006) subjected to long-term treatment with EGFR TKIs, including gefitinib, erlotinib, afatinib, and osimertinib. Gene expression profiles were recorded by microarray (GSE80344 and GSE106765) or RNAseq (GSE103350 and GSE95558), which were used to calculate the resistance-related score.

For each experiment, genes were ranked in descending order based on their TKI resistance-related scores. Finally, we adopted the discounted rating system [16], which was successfully used to prioritize disease candidate genes based on various data sources, to combine the 15 rankings into a single score (Appendix A).

A total of 26,423 genes were scored by integrating the 15 rankings. Gene Ontology (GO) enrichment analysis of the top 100 TKI resistance-related genes revealed the putative mechanisms of TKI resistance (Figure 1A). A few genes, such as protein kinase B (*AKT*), fibroblast growth factor (*FGF*), and transforming growth factor β (*TGFβ*), were involved in cell growth signaling and in epithelial–mesenchymal transition; these pathways have been reported to contribute to TKI resistance. Notably, a subset of genes is related to leukocyte activation and migration, suggesting that TKI-resistant cells modulate the immune microenvironment. Interestingly, *IGFBP7* was ranked as the top TKI resistance-related gene (Figure 1B). Integrative analysis revealed the impact of *IGFBP7* expression on TKI-resistance.

We also explored genetic alterations in the *IGFBP7* locus via cBioPortal of Cancer Genomics [17,18]. *IGFBP7* was altered in 16 samples (1%) from among 1530 profiled samples in seven datasets (Figure 1C). The genetic alterations in *IGFBP7* included 12 amplifications, three missense mutations, and one truncating mutation (Figure 1D,E).

Additionally, *IGFBP5* was listed as a highly raked candidate target gene. It was unknown whether a correlation exists in gene expression between *IGFBP7* and *IGFBP5*. We obtained the gene expression datasets for lung cancer cell lines from the Cancer Cell Line Encyclopedia and lung adenocarcinoma tissues from The Cancer Genome Atlas–Lung Adenocarcinoma Disease Type. There was no expression correlation (*r* = −0.16) between *IGFBP7* and *IGFBP5* in the cell line dataset, while *IGFBP5* expression was slightly correlated with *IGFBP7* in the tissue dataset (*r* = 0.26) (Appendix A).

### 2.2. IGFBP7 Was Significantly Elevated in EGFR-TKI-Resistant Cells

To confirm the impact of *IGFBP7* expression on TKI resistance, a cDNA microarray was used. PC9/gef, HCC827/gef, and HCC4006/ER cells were selected from TKI-sensitive parental cells (PC9, HCC827 and HCC4006) that had been continuously exposed to increasing concentrations of the EGFR TKIs gefitinib (gef) or erlotinib (ER). Cell viability determined using the 3-(4,5-dimethylthiazol-2-yl)-5-(3-carboxymethoxyphenyl)-2-(4-sulfophenyl)-2H-tetrazolium (MTS) assay revealed that PC9, HCC827, and HCC4006 cells were sensitive to gefitinib with half maximal inhibitory concentration (IC_50_) values of less than 0.1 μM after treatment for 96 h, while PC9/gef, HCC827/gef, and HCC4006/ER cells were resistant to gefitinib with IC_50_ values greater than 5 μM (Appendix A). The three EGFR-TKI-resistant sublines exhibited cross-resistance to other EGFR-TKIs, including afatinib (second-generation EGFR-TKI). These three EGFR-TKI-resistant cells contained neither T790M *EGFR* mutations nor *MET* amplifications.

The Affymetrix GeneChip Human Genome U133 Plus 2.0 Array (54,675 probes) was used to identify candidate genes exhibiting differential expression between EGFR-TKI-sensitive (PC9 and HCC827) and acquired EGFR-TKI-resistant cell lines (PC9/gef and HCC827/gef). These analyses revealed 50 candidate genes that were significantly altered (35 up-regulated and 15 down-regulated) in EGFR-TKI-resistant cell lines (Figure 2A). The top six genes showing differential expression (*IGFBP7*, Sperm protein associated with the nucleus, X-linked, family member A1 (*SPANXA1*), H-RAS-like suppressor (*HRASLS*), Ring finger protein 182 (*RNF182*), Sperm protein associated with the nucleus, X chromosome, family member C (*SPANXAC*), and Lymphocyte antigen 6 family member D (*Ly6D*)) were validated by quantitative reverse transcription (RT)-PCR. *IGFBP7* was expressed at significantly higher levels in EGFR-TKI-resistant cell lines compared to in parental controls (PC9/gef: 6.6-fold, HCC827/gef: 8.2-fold, and HCC4006/ER: 3.3-fold; Figure 2B). Western blotting analysis also indicated that the protein expression of IGFBP7 was up-regulated in EGFR-TKI-resistant cell lines compared to in EGFR-TKI-sensitive cells (Figure 2C). Gene expression analysis results were consistent with those of the integrative analysis of the public datasets.

### 2.3. Higher IGFBP7 Levels in Malignant Pleural Effusion of Lung Adenocarcinoma After Acquired Resistance to EGFR-TKIs

To confirm the change in *IGFBP7* after acquired resistance to EGFR-TKIs, we collected 24 malignant pleural effusions of *EGFR*-mutant lung adenocarcinoma for *IGFBP7* mRNA analysis. The mean mRNA value of *IGFBP7* expression in cancer cells isolated from treatment-naïve pleural effusions (*n* = 8) (0.114 ± 0.059) was significantly lower than that after acquired resistance to EGFR-TKI treatment (*n* = 16) (1.18 ± 0.412; *p* = 0.0212, by Student’s *t*-test). (Figure 2D). 

### 2.4. Suppression of IGFBP7 in EGFR-TKI-Resistant Cells Enhanced Gefitinib-Induced Cell Death

To examine the role of *IGFBP7* in the resistance to EGFR-TKIs, *IGFBP7*-specific siRNA (si-IGFBP7-1 and si-IGFBP7-4) against *IGFBP7* was used to knock down *IGFBP7* in EGFR-TKI-resistant cells (PC9/gef, HCC4006/ER and HCC827/gef). We found that transfection with *IGFBP7*-specific siRNAs suppressed *IGFBP7* expression and recovered EGFR-TKI sensitivity in EGFR-TKI-resistant cells (PC9/gef, HCC4006/ER, and HCC827/gef; Figure 3A,B and Appendix A). The percentage of apoptotic cells, which was quantified as Annexin-V-positive cells, was significantly increased in *IGFBP7*-suppressed cells compared to in control cells (PC9/gef-si-scramble and HCC4006/ER-si-scramble) following exposure to gefitinib and afatinib (Figure 3C and Appendix AA).

Gefitinib treatment clearly induced cleavage of caspase-3, caspase-7, poly-(ADP-ribose) polymerase (PARP), and B-cell lymphoma 2 interacting mediator of cell death (BIM) in *IGFBP7*-knockdown-PC9/gef cells (PC9/gef-si-IGFBP7-1 and PC9/gef-si-IGFBP7-4) compared to in PC9/gef-si-scramble cells (Figure 3D). *IGFBP7*-knockdown-HCC4006/ER (HCC4006/ER-si-IGFBP7-1 and HCC4006/ER-si-IGFBP7-4) and *IGFBP7*-knockdown-HCC827/gef cells (HCC827/ER-si-IGFBP7-1) also showed increased levels of apoptosis markers compared to in HCC4006/ER-si-scramble and HCC827/gef-si-scramble cells, respectively (Appendix A). Furthermore, recombinant IGFBP7 reduced the levels of gefitinib-induced cleaved caspase-3 and caspase-7 in *IGFBP7*-knockdown-PC9/gef cells (Figure 3E). These results revealed that knockdown of *IGFBP7* expression restored EGFR-TKI sensitivity in EGFR-TKI-resistant cells by increasing apoptosis. *IGFBP7* plays a unique role in EGFR-TKI resistance.

### 2.5. Enhanced IGFBP7 Expression Prevents EGFR-TKI-Induced Apoptosis

We established *IGFBP7*-overexpressing cells from EGFR-TKI-sensitive PC9 cells. *IGFBP7*-transfected cells (PC9-IGFBP7) expressed higher levels of *IGFBP7* mRNA and protein than control cells (PC9-mock; Figure 4A,B). After gefitinib treatment, the percentage of apoptotic cells was decreased in PC9-IGFBP7 cells compared to in PC9-mock cells (Figure 4C,D). Apoptosis markers were significantly induced after treating the PC9 cells for 24 h with 0.025 μM gefitinib. This gefitinib concentration was used to explore the inhibition condition of apoptosis markers in PC9-IGFBP7 cells. Western blotting showed that IGFBP7 inhibits gefitinib-induced apoptosis (activation of caspase-7 and caspase-9) in PC9-IGFBP7 cells (Figure 4E). BIM expression following gefitinib treatment was also decreased in PC9-IGFBP7 cells compared to in PC9-mock cells; thus, gefitinib-induced BIM up-regulation was abolished by IGFBP7 (Figure 4E). *IGFBP7* overexpression may suppress caspase activity and BIM expression to protect PC9 cells from gefitinib-induced apoptosis.

### 2.6. Suppression of IGFBP7 Attenuates Phosphorylation of IGF-1R and Downstream AKT

Recent reports showed that the insulin-like growth factor-1 (IGF-1) pathway may be involved in EGFR-TKI resistance [19,20]. Activation of IGF-1 receptor (IGF-1R) signaling confers resistance to afatinib in *EGFR* T790M-mutant lung cancer cells [21]. In this study, we showed that *IGFBP-7*-knockdown cells (including PC9/gef-si-IGFBP7-1, PC9/gef-si-IGFBP7-4, HCC4006/ER-si-IGFBP7-1, and HCC4006/ER-si-IGFBP7-4 cells) had lower phosphorylation levels of IGF-1R and AKT than control cells (Figure 5A,B). Suppression of IGFBP7 in PC9/gef and HCC4006/ER cells reduced the phosphorylation of IGF-1R and AKT, but not extracellular signal-regulated kinase (ERK) phosphorylation. This indicates that knockdown of *IGFBP-7* reversed the EGFR-TKI resistance mechanism by inhibiting the IGF-1R pathway.

### 2.7. Higher IGFBP7 Level Is Associated with Shorter Progression-Free Survival of EGFR-TKI-Treated Lung Cancer Patients

We collected a cohort (102 lung adenocarcinoma specimens) for IGFBP7 immunohistochemical (IHC) staining (Figure 6A). All patients had *EGFR*-mutant lung adenocarcinoma, and were administered EGFR-TKIs (91 gefitinib and 11 erlotinib) as the first-line treatment. Using the IGFBP7 IHC criteria in the Methods section, 53 patients had IGFBP7 IHC-positive tumors and 49 had IGFBP7 IHC-negative tumors. There was no significant difference in clinical characteristics between the two IGFBP7 groups (Appendix A). Patients with IGFBP7 IHC-negative tumors had longer progression-free survival (PFS) following EGFR-TKI treatment than IGFBP7 IHC-positive patients (median 19.9 months vs. 13.9 months; *p* = 0.028) (Figure 6B). However, there was no significant difference in overall survival (OS) after gefitinib treatment between patients with negative and positive IGFBP-7 IHC (median 52.8 months vs. not reached; *p* = 0.540) (Figure 6C).

We consecutively collected 75 peripheral blood samples from patients diagnosed as having *EGFR*-mutant lung adenocarcinoma who were administered gefitinib or erlotinib as the first-line treatment. The clinical characteristics of the patients are listed in Appendix A. The serum IGFBP7 level was measured by enzyme-linked immunosorbent assay (ELISA). We divided the patients into high and low serum level groups based on the median value of IGFBP7 (544.96 ng/mL; range: 87.86–3199.32 ng/mL). The low IGFBP7 group showed a longer median PFS with EGFR-TKI treatment than the high IGFBP7 group (13.4 months vs. 8.1 months; *p* = 0.036; Figure 6D). There was also no significant difference in OS after gefitinib treatment between the low and high IGFBP7 groups (median 23.8 months vs. 16.4 months; *p* = 0.764) (Figure 6E).

## 3. Discussion

IGFBPs are circulating proteins that are traditionally known as carrier proteins that modulate the activity of IGFs. However, IGFBPs can bind to specific cell membrane receptors and attach to the cell surface or extracellular matrix [22]. The family of IGFBPs has been shown to have multiple and complex functions including in cell proliferation, motility, and tissue remodeling, which can exert either IGF-dependent or IGF-independent mechanisms [22,23]. As shown in recent studies, elevated *IGFBP2* expression and activated *FAK* have been causally associated with dasatinib resistance in lung cancer cells, and administration of a combination of dasatinib with an FAK inhibitor can overcome this resistance [24]. Both *IGFBP2* and *FAK* can be used as biomarkers for identifying dasatinib responders among lung cancer patients [24].

The IGFBP family has been reported to be involved in EGFR-TKI resistance. *IGFBP3* up-regulation in lung cancer cells leads to increased *MET* expression via Smad2/3 activation and results in EGFR- and Met-TKI dual resistance [25]. Other studies showed that increased *IGFBP3* was observed in PC9 afatinib-resistant cells (AFR2 cells) compared to in parental PC-9 cells [26]. This study indicated that increased *IGFBP3* expression resulted in enhanced *MET* expression and ERK1/2 phosphorylation. However, IGFBP3 was observed to be inversely correlated with TKI resistance [19]. These results suggest that the function of IGFBP3 is unclear in EGFR-TKI resistance, which may be influenced by the cell context. In this study, we found that IGFBP7 contributes to resistance to EGFR-TKIs and showed that enhanced IGFBP7 expression promoted resistance to EGFR-TKI in lung cancer cells by mediating the IGF-1R pathway. Our findings suggest that co-targeting of EGFR and IGFBP7 is an effective strategy for treating *EGFR*-mutant lung cancer.

In this study, analysis of public Gene Expression Omnibus (GEO) datasets and our two-paired cell line screening showed that *IGFBP7* was up-regulated in EGFR-TKI-resistant cells. IGFBP7, a secreted 31-kDa protein, is related to a member of the IGFBP family [27]. Unlike the other family members (IGFBP1-6), this protein exhibits low affinity for IGF but high and specific affinity for insulin, as well as conferring a level of regulation to the IGF signaling system [27,28]. Varied IGFBP7 expression patterns have been reported in different tumor types [29,30]. Methylation-dependent silencing of *IGFBP7* was associated with unfavorable outcomes in colorectal, breast, and pancreatic cancers [29,31,32,33,34,35]. *IGFBP7* was reported to act as a tumor suppressor by regulating cell proliferation, invasion, and angiogenesis [23]. *IGFBP7* overexpression is also being evaluated as a potential cancer biomarker [31,32,33,34,35]. *IGFBP7* has been detected in invasive prostate neoplasms compared to in normal secretory or benign prostatic hyperplasia [36]. *IGFBP7* expression was also linked to a poor prognosis in multiple myeloma, colorectal cancer, esophageal adenocarcinoma, and head and neck squamous cell carcinomas [37,38,39,40]. In NSCLC, IGFBP7 may be involved in lymphangiogenesis and is associated with metastatic clinicopathological features [41]. A high serum level of IGFBP7 was correlated with a positive lymph node status in lung cancer [42]. The role of IGFBP7 in solid tumors is inconsistent, and recent studies revealed a more complex picture of IGFBP7 in different entities [43,44]. However, no previous studies have examined the correlation between IGFBP7 and acquired EGFR-TKI resistance in lung cancer.

In the present study, we found that IGFBP7 affected EGFR-TKI resistance by using gain- and loss-of-function approaches. IGFBP7 expression is highly associated with EGFR-TKI resistance in lung cancer cells. A reduction of *IGFBP7* expression in EGFR-TKI-resistant cells (PC9/gef and HCC4006/ER cells) was shown to restore gefitinib-induced cell death by up-regulating BIM and activating caspases. While enhanced *IGFBP7* expression promoted resistance to EGFR-TKIs in lung cancer cells, IGF-1R activation has been reported to confer acquired resistance to EGFR-TKIs [19,20,21]. Our results also showed that restoring EGFR-TKI sensitivity by suppressing IGFBP7 in EGFR-TKI-resistant cells attenuated the phosphorylation of IGF-1R and downstream AKT. Thus, IGFBP7 may be a novel target for blocking IGF-1R signaling in lung cancer cells. However, the treatment efficacy of IGF-1R inhibition must be further evaluated and confirmed.

## 4. Materials and Methods 

### 4.1. Data Integration for TKI-Associated Gene Prioritization and Candidate Gene Alteration Exploration

We collected gene expression profiles of EGFR-TKI-sensitive and resistant human lung cancer cell lines from the GEO and applied the discounted rating system to prioritize TKI-associated genes (Appendix A) [16].

Additionally, we used cBioPortal to explore gene alterations in the candidate gene locus, including copy number variation and mutations [17,18]. We queried lung adenocarcinoma samples in 7 different datasets [45,46,47,48].

### 4.2. Bioinformatics Analysis

Gene ontology (GO) enrichment analysis was performed using the R/Bioconductor clusterProfiler package [49]. The enrichment map was constructed as described by Merico et al., with overlap scores > 0.5, and visualized with Cytoscape (v 3.6.1) [50].

### 4.3. Cell Culture, Small Interfering RNA (siRNA), Recombinant Protein and Drugs

The human lung adenocarcinoma cell line PC9 and derivative PC9/gef clones were gifts from Dr. James Chih-Hsin Yang (Department of Oncology, National Taiwan University Hospital, Taipei City, Taiwan). HCC827 and HCC4006 cells were purchased from the American Type Culture Collection (Manassas, VA, USA). PC9, HCC827, and HCC4006 lung cancer cells all contain *EGFR* mutations with a deletion in exon 19 and are sensitive to EGFR-TKIs. PC9/gef and HCC827/gef cells were selected from parental cells (PC9 and HCC827) that had been continuously exposed to increasing concentrations of gefitinib [11,12,51]. HCC4006/ER was selected from HCC4006 after long-term culture with increasing concentrations of erlotinib. PC9, PC9/gef, HCC827, HCC827/gef, HCC4006, and HCC4006/ER cells were cultured in Roswell Park Memorial Institute (RPMI)-1640 medium supplemented with 10% heat-inactivated fetal bovine serum and antibiotics (100 U/mL penicillin and 100 U/mL streptomycin) at 37 °C in a 5% CO_2_/95% air atmosphere. *IGFBP7* was silenced in PC9/gef and HCC4006/ER cells using specific IGFBP7 siRNA duplexes (Invitrogen, Carlsbad, CA, USA). Recombinant human IGFBP7 protein was purchased from R&D Systems (Minneapolis, MN, USA). Gefitinib, erlotinib, and afatinib were purchased from Selleckchem (Houston, TX, USA) and prepared in dimethyl sulfoxide to obtain a stock solution of 10 mM.

### 4.4. Gene Expression Microarray Profiling

Total RNAs were prepared from EGFR-TKI-sensitive cell lines (PC9 and HCC827) and cell lines with acquired resistance to EGFR-TKIs (PC9/gef and HCC827/gef), with three biological replicates for each group. RNAs were extracted, labeled, and hybridized to the Affymetrix GeneChip Human Genome U133 Plus 2.0 Array following the manufacturer’s protocol (Santa Clara, CA, USA). The obtained data were submitted to GEO (GSE122005).

### 4.5. Establishment of IGFBP7-Expressing Stable Cell Lines 

*IGFBP7* was overexpressed in PC9 cells by infecting the cells with lentiviruses containing the entire human *IGFBP7* coding region and prepared using the ViraPower Lentiviral Expression System (Invitrogen) as described by the manufacturer. A stable transfectant prepared by applying 500 μg/mL gentamicin (G418; Invitrogen) for 2 weeks was selected. PC9/IGFBP7 cells were selected as representative IGFBP7 overexpression clones. Mock transfected cells (PC9/mock) were used in bulk as a control.

### 4.6. Quantitative Reverse Transcriptase-Polymerase Chain Reaction (RT-PCR)

Total RNA was extracted from the cell lines using TRIzol reagent (Invitrogen) according to the manufacturer’s protocol, and RNA was reverse-transcribed using random primers and reverse transcriptase following the manufacturer’s protocol (Invitrogen). Quantitative RT-PCR was performed in the Real-Time PCR System 7900 (Applied Biosystems, Foster City, CA, USA). The primer sequences are listed in Appendix A. The relative gene expression of each sample was determined using the formula 2^(−Δ*C*t)^ = 2^(*C*t (TBP) − *C*t (IGFBP7))^, which reflects the target gene expression normalized to TATA-box binding protein (TBP) levels.

### 4.7. ELISA

The serum IGFBP7 levels in patients with lung cancer were analyzed using a commercially available ELISA kit (USCN Life Science, Inc., Wuhan, China) according to the manufacturer’s protocol.

### 4.8. Cytotoxicity Assay

We used the colorimetric MTS assay (CellTiter 96 AQueous One Solution Cell Proliferation Assay Kit; Promega, Madison, WI, USA) to determine the number of viable cells. Absorbance at 490 nm was recorded using a VICTOR3 multilabel reader (PerkinElmer, Waltham, MA, USA).

### 4.9. Apoptosis Assay

The Annexin V-FITC Apoptosis Detection Kit (BD Biosciences, Franklin Lakes, NJ, USA) was used to detect apoptosis. Briefly, the cells were trypsinized and washed twice with ice-cold phosphate-buffered saline. The cell pellet was resuspended and incubated in Annexin V binding buffer containing fluorescein isothiocyanate-conjugated Annexin V (1 μg/mL) and propidium iodide (50 μg/mL) for 15 min at room temperature. Analyses were performed using a Cytomic FC500 flow cytometer (Beckman Coulter, Brea, CA, USA).

### 4.10. Western Blotting

Cell lysates were prepared and quantified using a Pierce BCA protein assay kit (Thermo Fisher Scientific, Waltham, MA, USA). Western blotting was conducted as described previously [11]. Antibodies are listed in Appendix A. Monoclonal anti-β-actin or anti-α-tubulin antibodies (Millipore, Billerica, MA, USA) against β-actin or α-tubulin proteins were used as a loading control.

### 4.11. Patients and Sample Collection 

Human lung adenocarcinoma samples were collected and archived under protocols approved by the Institutional Review Board (IRB) of National Taiwan University Hospital (NTUH) (REC No. 201101012RC, 8 February 2011). All patients provided informed consent for participation before the collection of information and samples. The study was conducted in accordance with the Declaration of Helsinki. Lung cancer was confirmed by pathologic or cytological diagnoses using tissues obtained from biopsy or aspiration. Lung adenocarcinoma was diagnosed according to the International Multidisciplinary Classification of Lung Adenocarcinoma criteria [52]. Disease stage was determined using the seventh edition of the International Association for the Study of Lung Cancer tumor-node-metastasis (TNM) staging system [53]. A complete lung cancer staging work-up was performed as a routine practice, and included bronchoscopy, computed tomography of the head, chest and abdomen, and whole-body bone scintigraphy. The clinical information of all patients, including age, sex, smoking history, and lung cancer stage, was recorded.

### 4.12. Malignant Pleural Effusion Isolation

Pleural effusions were consecutively collected from patients who underwent thoracentesis in the chest ultrasonography examination room of NTUH. All patients had signed an informed consent form for future molecular analyses before thoracentesis was performed. The pleural fluids of patients were acquired aseptically in vacuum bottles by thoracentesis. After centrifugation, cancer cells were isolated and cultured [11]. The media were replaced every 2–3 days and cells were harvested after 10 days.

### 4.13. Tumor Specimens for Immunohistochemical Staining (IHC)

Four-micrometer sections were cut from the paraffin-embedded tissue samples of the lung adenocarcinomas of patients administered EGFR-TKIs as the first-line treatment. Antigen was retrieved using AR-10 Solution (Biogenex San Ramon, CA, USA). The monoclonal antibody IGFBP7 (Cat. Number: HPA002196, Sigma-Aldrich, St. Louis, MO, USA) was used (dilution: 1:150 for both antibodies) and the slides were incubated overnight at 4 °C. A diaminobenzidine (BioGenex) detection kit was used. The slides were then counter-stained with hematoxylin. Two observers evaluated the staining results independently and differences in interpretation were resolved by consensus. The intensity was scored from 0 to 3+ and defined as follows: 0, no staining; 1+, weak staining; 2+, moderate staining; 3+, strong staining based on the staining score [54]. Positive staining of IGFBP7 IHC was defined as moderate staining (2+) in more than 10% of the tissue.

### 4.14. Peripheral Blood Sample Collection

We consecutively collected peripheral blood samples from patients who were diagnosed as having lung adenocarcinoma harboring *EGFR* mutations. The enrolled patients were administered EGFR-TKIs as the first-line treatment. All blood samples were obtained before EGFR-TKI treatment. This study was approved by the IRB of NTUH, and all patients gave written informed consent before blood sampling.

### 4.15. Statistical analysis

Student’s *t*-test was used to compare the means of two groups. The log-rank test was used to compare the PFS and OS after treatment with EGFR-TKI in the two groups. Two-sided *p*-values less than 0.05 were considered significant. All analyses were performed using SPSS software (version 15.0 for Windows; SPSS, Inc., Chicago, IL, USA).

## 5. Conclusions

We found that IGFBP7 expression was significantly elevated in EGFR-TKI-resistant cells, and that high plasma IGFBP7 levels or IGFBP7 IHC-positive tumors in patients with lung cancer were correlated with shorter PFS when EGFR-TKI was used as the first-line treatment. Serum IGFBP7 levels before EGFR-TKI treatment could predict PFS with EGFR-TKI treatment in patients with *EGFR*-mutant lung adenocarcinoma. The clinical picture was similar to that observed in vitro. Early detection of IGFBP7 may have implications for more aggressive follow-up. Our study reveals the important role of IGFBP7 and suggests that it can be used as a potential therapeutic target for overcoming EGFR-TKI resistance. This may mark the beginning of a new era in overcoming resistance to EGFR-TKIs and suggests a new treatment strategy for lung cancer.

## Figures and Tables

**Figure 1 cancers-11-00036-f001:**
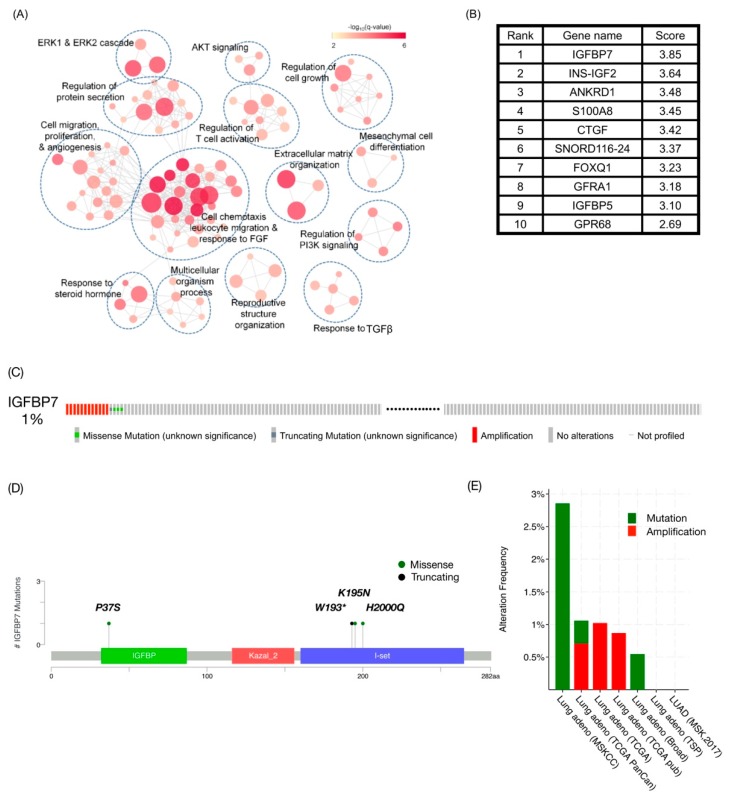
Integrative analysis to identify tyrosine kinase inhibitor (TKI) resistance-related genes. (**A**) Enrichment map of the top 100 TKI resistance-related genes. (**B**) Top 10 TKI resistance-related genes. The nodes represent Gene Ontology (GO) terms, and node colors and size indicate the enrichment significance and number of GO-associated genes. The edges indicate the large amount of overlap between GO-associated genes. (**C**) The oncoprint revealed that 1% (16 altered/1530 profiled) of Lung adenocarcinoma (LUAD) patients had alterations in insulin-like growth factor binding protein 7 (IGFBP7) in the lung adenocarcinoma cohort from cBioPortal of Cancer Genomics. (**D**) Three missense mutations and one truncating mutation in the IGFBP7 locus (**E**) Alteration frequency in different datasets. MSK: Memorial Sloan Kettering Cancer Center; TSP: The Tumor Sequencing Project; TCGA: The Cancer Genome Atlas; MSKCC: Memorial Sloan Kettering Cancer Center; ERK: extracellular-signal-regulated kinase; AKT: protein-kinase B; PI3K: phosphoinositide 3-kinase.

**Figure 2 cancers-11-00036-f002:**
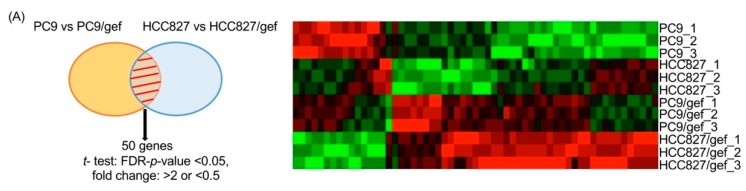
IGFBP7 was up-regulated in epidermal growth factor receptor (EGFR)-TKI-resistant cells. (**A**) Affymetrix GeneChip Arrays were performed to identify differentially expressed genes between EGFR-TKI-sensitive (PC9 and HCC827) and acquired EGFR-TKIs-resistant cell lines (PC9/gefitinib (gef) and HCC827/gef). (**B**) The top six differentially expressed genes are listed and were validated by quantitative RT-PCR. (The levels of *SPANXC* were too low to be detected). (**C**) IGFBP7 protein expression was detected by Western blot analysis. EGFR-TKI-sensitive cells (PC9, HCC827, and HCC4006); EGFR-TKI-resistant cells (PC9/gef, HCC827/gef and HCC4006/erlotinib (ER)). (**D**) *IGFBP7* mRNA levels were detected by quantitative RT-PCR. The mean of the *IGFBP7* expression level was higher in malignant pleural effusions of patients with lung adenocarcinoma after acquired resistance to EGFR-TKIs than in that of treatment-naïve patients. The scatter dot plot graph represents *IGFBP7* expression, and the error bars represent standard errors of the mean. (* *p* = 0.021).

**Figure 3 cancers-11-00036-f003:**
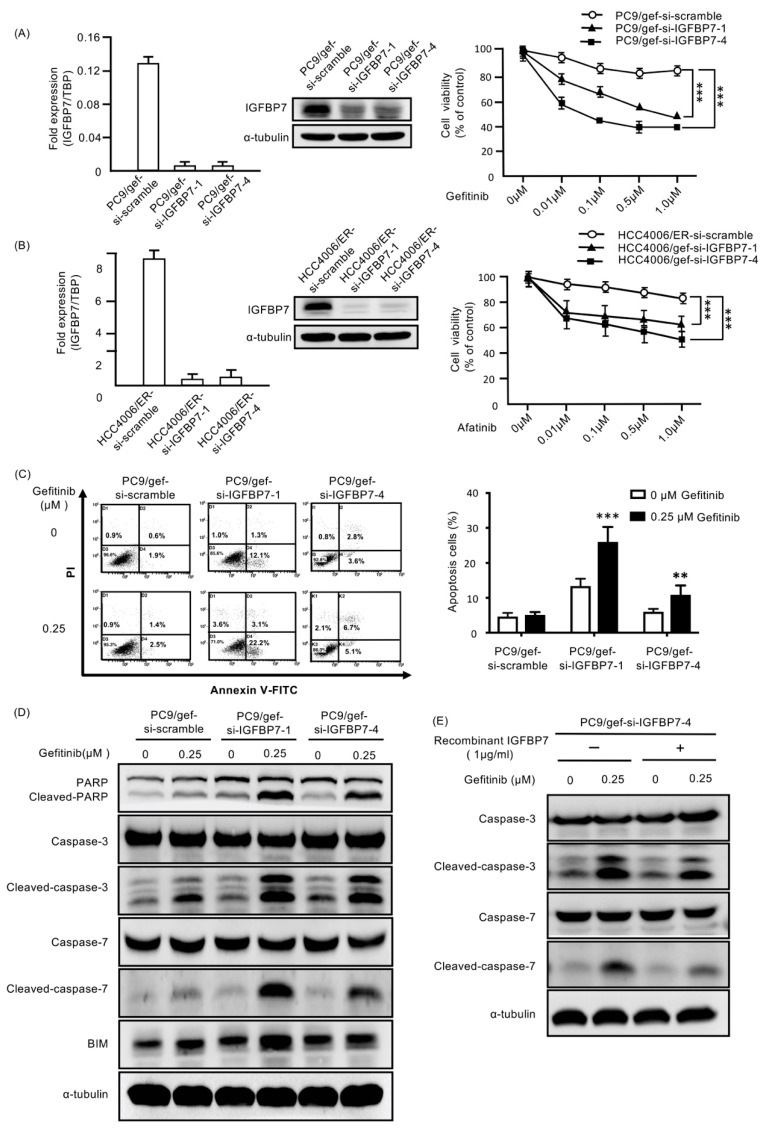
Knockdown of *IGFBP7* expression reversed EGFR-TKI resistance in PC9/gef, HCC4006/ER, and HCC827/gef cells by enhancing EGFR-TKI-induced caspase activity and BIM expression (**A**) PC9/gef cells and (**B**) HCC4006/ER cells were transfected with *IGFBP7* small interfering RNAs (siRNAs; si-IGFBP7-1 and si-IGFBP7-4) or scramble siRNA (si-scramble). The effect of siRNAs was evaluated by quantitative RT-PCR and Western blot analysis (left). Cellular viability of si-scramble and si-IGFBP7 transfectants was determined following treatment with various doses of gefitinib for 96 h in MTS assays (right). Error bars show the standard deviations for *n* = 3 independent experiments (*** *p* < 0.001). (**C**) The percentage of apoptotic cells was quantified after treatment with gefitinib (0.25 μM). The columns are the means of three independent experiments. Error bars show the standard deviations (** *p* < 0.01; *** *p* < 0.001). (**D**) PC9/gef was exposed to 0.25 μM of gefitinib for 24 h. Next, apoptosis markers, including cleaved poly-(ADP-ribose) polymerase (PARP), caspase-3, caspase-7 and BIM, were assayed by Western blotting. (**E**) *IGFBP7*-knockdown-PC9/gef cells were treated with recombinant IGFBP7 (1 μg/mL), and then cleaved-caspase-3 and cleaved-caspase-7 were evaluated by Western blotting. (PARP: Poly ADP ribose polymerase; TBP: TATA-binding protein; PI: Propidium iodide; BIM: B-cell lymphoma 2 interacting mediator of cell death; MTS: 3-(4,5-dimethylthiazol-2-yl)-5-(3-carboxymethoxyphenyl)-2-(4-sulfophenyl)-2H-tetrazolium).

**Figure 4 cancers-11-00036-f004:**
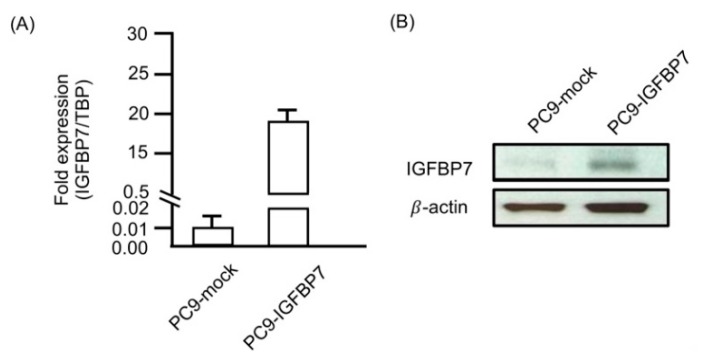
Overexpression of IGFBP7 protects PC9 cells from gefitinib-induced apoptosis. The expression of IGFBP7 in stable transfectants (PC9-mock and PC9-IGFBP7) was evaluated by (**A**) Quantitative RT-PCR and (**B**) Western blot analysis. (**C**) After 48 h of treatment with gefitinib (0 nM, 50 nM and 300 nM), the cells were stained with Annexin V–fluorescein isothiocyanate (FITC) and propidium iodide (PI). The percentage of apoptotic cells was evaluated by flow cytometry. (**D**) The columns show the means of three independent experiments. Error bars show standard deviations (* *p* < 0.05; ** *p* < 0.01). (**E**) After 24 h of treatment with 0 nM, 10 nM and 25 nM gefitinib, the apoptotic markers caspase-7, caspase-9, and BIM were evaluated by Western blot analysis.

**Figure 5 cancers-11-00036-f005:**
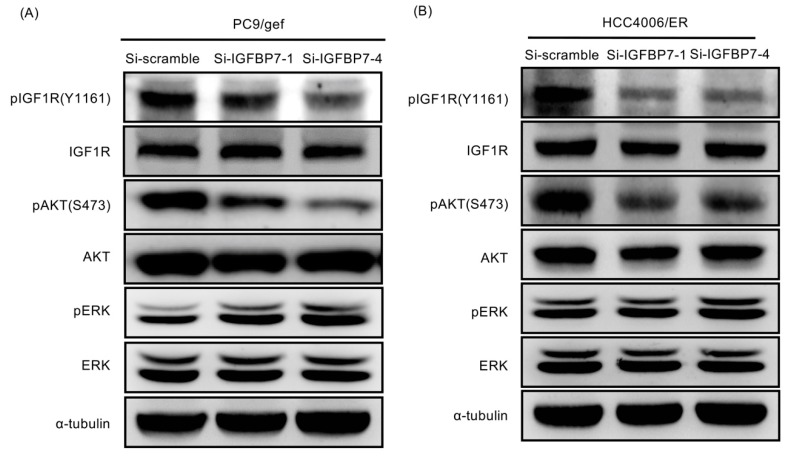
Suppression of IGFBP7 reduced phosphorylation of IGF-1R and downstream AKT (**A**) PC9/gef and (**B**) HCC4006/ER were transfected with si-IGFBP7-1, si-IGFBP7-4, or scramble siRNA (si-scramble) for 48 h. Western blotting showed that *IGFBP7*-knockdown cell lines decreased the phosphorylation of IGF-1R and AKT, except for ERK phosphorylation.

**Figure 6 cancers-11-00036-f006:**
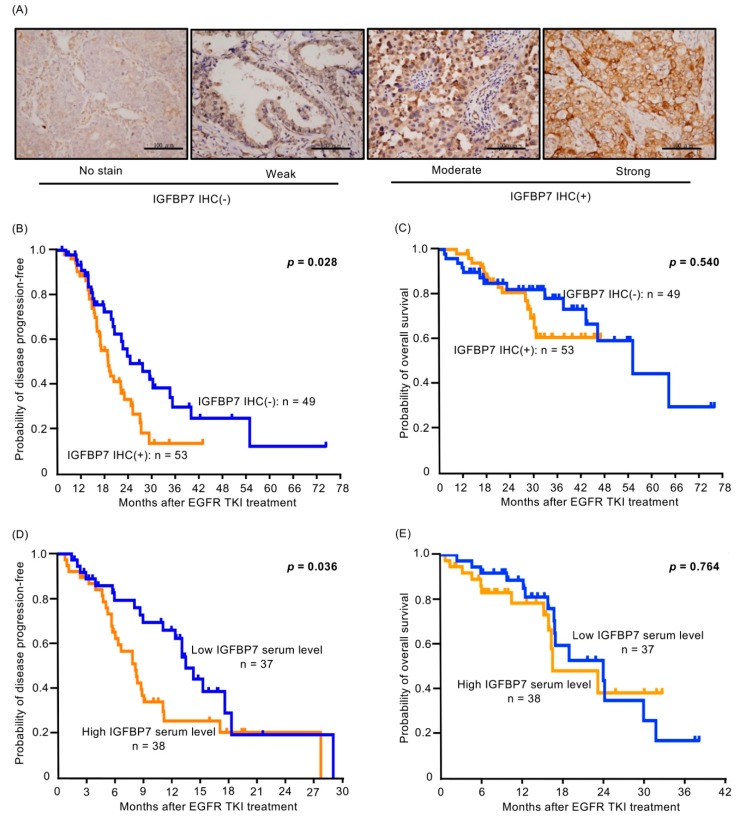
(**A**) Representative immunohistochemical (IHC) staining results for IGFBP7 in samples from patients with lung adenocarcinoma with an EGFR mutation who were administered gefitinib as the first-line treatment. Scale Bar: 100 µm. (**B**) Patients with IGFBP7 IHC-negative lung adenocarcinoma (blue line) had longer progression-free survival (PFS) with EGFR-TKIs than those with IHC-positive lung adenocarcinoma (yellow line) (median 19.9 months vs. 13.9 months; *p* = 0.028). (**C**) No significant difference was observed in overall survival (OS) after gefitinib treatment between patients with negative and positive IGFBP-7 IHC (median 52.8 months vs. not reached; *p* = 0.540) (**D**) Patients with low serum IGFBP7 levels (blue line) had a longer median PFS with EGFR-TKIs than those with high serum IGFBP7 levels (yellow line) (13.4 months vs. 8.1 months; *p* = 0.036). (**E**) No significant difference was observed in OS after gefitinib treatment between patients with low and high IGFBP-7 serum level (median 23.8 months vs. 26.4 months; *p* = 0.764).

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
