# Peer review of "IGFBP7 Drives Resistance to Epidermal Growth Factor Receptor Tyrosine Kinase Inhibition in Lung Cancer"

_cancers, 2019, doi:10.3390/cancers11010036_

Round 1

Reviewer 1 Report

I found this study very interesting in order to better understand the molecular mechanisms underline TKI resistance and to ameliorate targeted therapeutic approach.

I have no concern about this paper and it is possible to publish in this form.

Author Response

  Thanks for your review.

Reviewer 2 Report

The authors provide extensive evidence that IGFBP7 might play a role in resistance of EGFR-Mutant NSCLC to EGFR-TKIs. Both in vitro data (including loss-of-function and gain-of-function approaches) and clinical data (including Progression-free survival (PFS) data) are provided. These data are certainly relevant, in particular since a gain of function in the IGFBP7 pathway might serve as therapeutic target.

There is only one Major point: It would be desirable to have the Overall survival data of the patients in figure 6B and 6C.

Author Response

1.        It would be desirable to have the Overall survival data of the patients in figure 6B and 6C.

R: The analysis result of overall survival (OS) was attached in revised manuscript.

è The OS of the patients was shown in Figure 6C and 6E, and the details were described in the paragraph of page 10 and the legend of Figure 6.

èLine 236-238: ” However, there was no significant difference in overall survival (OS) after gefitinib treatment between patients with negative and positive IGFBP-7 IHC (median 52.8 months vs. not reached; p = 0.540) (Figure 6C).”

èLine245-247: “There was also no significant difference in OS after gefitinib treatment between the low and high IGFBP7 groups (median 23.8 vs. 16.4 months; p = 0.764) (Figure 6E).”

èPage 11 the legend of Figure 6: “(C) No significant difference was observed in OS after gefitinib treatment between patients with negative and positive IGFBP-7 IHC (median 52.8 vs. not reached; p = 0.540) (D)”and “(E) No significant difference was observed in OS after gefitinib treatment between patients with low and high IGFBP-7 serum level (median 23.8 vs. 26.4 months; p = 0.764).”

è In Line 416, the statistic of OS was added in the 4.15 of Method: “The log-rank test was used to compare the PFS and OS after treatment with EGFR-TKI in the two groups.”

Reviewer 3 Report

Wu review

This is a well-prepared and well-conducted study investigating the role of IGFBP7 in evolved resistance to tyrosine kinase inhibition (TKI) in EGFR-mutant lung cancer. TKI resistance is a notable clinical problem, to the point that newer generation EGFR-TKIs also target known secondary mutations such as T790M, that arise from tyrosine kinase inhibition therapy. However, there is emerging literature on second-site changes outside of the EGFR locus that lead to TKI resistance. IGFBP7 has been well-researched in a variety of fields, including in several GI cancers, and while findings are mixed, upregulation of IGFBP7 seems to be associated with more aggressive cancer and worse outcome. Moreover, the IGFBP family in general has been implicated in a wide variety of cancers, including IGFBP3 which may be inversely correlated with TKI resistance.

The study begins by requerying a broad swath of experiments with publicly available data on the Gene Expression Omnibus. After ranking, IGFBP7 was the most frequently differentially expressed gene between TKI-sensitive and TKI-resistant cells. This was then confirmed by cDNA microarray, including both cells with innate and acquired resistance to TKIs. A similar finding was noted in cancer cells isolated from malignant pleural effusions isolated from patients with treatment naïve EGFR-activated cancer and patients with acquired resistance to TKIs. Using siRNA knockdown, the authors then go on to show necessity of IGFBP7 upregulation in lung cancer cell lines with TKI resistance, and then used overexpression to show sufficiency for IGFBP7 overexpression in inducing TKI resistance in sensitive cells. This mechanism seems to work through apoptosis, with IGFBP7 knockdown inducing caspase cleavage and overexpression protecting against it. The authors conclude that, by Western blotting, IGFBP7 induces Akt and IGB1R phosphorylation. They then demonstrate immunohistochemistry and plasma levels of IFGB7 are correlated with progression free survival.

The manuscript represents a significant amount of work, with clear conclusions. The authors do a commendable job supporting their hypotheses and I believe this paper should be accepted with only minor revisions.

1.    It would be useful to briefly describe the GEO datasets used for initial analysis. Do these include the gene expression for the cell lines used in the rest of the paper?

2.    Is there an explanation for the 10x difference in gefitinib doses used in the PC9 cells with siRNA knockdown and overexpression?

3.    Did the authors query TCGA or other databases for copy number variation or mutation in the IGFBP7 locus?

4.    Was IGFBP7 expression correlated with other IGFBPs? IGFBP5 is listed as a highly ranked target in Figure 1B.  

Author Response

1.        It would be useful to briefly describe the GEO datasets used for initial analysis. Do these include the gene expression for the cell lines used in the rest of the paper?

R: We briefly described the enrolled GEO datasets and revised the Supplemental Table S1 for the detail of cell lines in different GEO datasets.

èWe added the description about these datasets in Line 77-82:

In each dataset, corresponding TKI-resistant cell lines were generated from TKI-sensitive cell lines (PC9, HCC827, and HCC4006) subjected to long-term treatment of EGFR TKIs, including with gefitinib, erlotinib, afatinib, and osimertinib. Gene expression profiles were recorded by microarray (GSE80344 and GSE106765) or RNAseq (GSE103350 and GSE95558), which were used to calculate the resistance-related score.

èWe revised the Supplemental Table S1.

    2.   Is there an explanation for the 10x difference in gefitinib doses used in the PC9       cells with siRNA knockdown and overexpression?

R: In this study, we used PC9/gef for IGFBP7-knockdown study and PC9 for IGFBP7-overexpression study. The IC50 of the two cell lines, EGFR TKI-sensitive and -resistance cells, are quite different (IC50 of gefitinib in PC9 = 0.04 μM, and IC50 of gefitinib in PC9/gef = 6.01 μM). Based on separate IC50 of gefitinib, two strategies were used in the study. We first knockdown IGFBP7 in PC9/gef cells to examine whether loss of IGFBP7 increased EGFR TKI-induced cell death. To explore whether suppressed IGFBP7 expression could recover EGFR-TKI sensitivity in PC9/gef, cell viability was exanimated under 0.01, 0.1, 0.5, and 1.0 μM of gefitinib (Fig 3A). The IGFBP7-knockdown-PC9/gef cells were exposed to 0.25 μM of gefitinib, and then the apoptosis markers were evaluated (Fig 3D). In contrast, PC9 cell line was used in conducting introduction of IGFBP7 and assess its effects on decreasing EGFR-TKI-induced apoptosis. The dose of 0.025μM was chosen to explore the inhibition condition of apoptosis markers in PC9/IGFBP7 cells because the dose is similar to the IC50 of PC9 cells.

 è We added the description in Line 195-198 of the results:“Apoptosis markers were significantly induced after treating the PC9 cells for 24 h with 0.025 μM gefitinib. This gefitinib concentration was used to explore the inhibition condition of apoptosis markers in PC9-IGFBP7 cells.”

3.    Did the authors query TCGA or other databases for copy number variation or mutation in the IGFBP7 locus?

R: To examine genetic alterations in IGFBP7, we used cBioPortal to query lung adenocarcinoma samples from 7 different datasets. The 7 dtatsets included Lung Adenocarcinoma (Broad, Cell 2012, Lung Adenocarcinoma (MSKCC 2015), Lung Adenocarcinoma (TCGA, Nature 2014), Lung Adenocarcinoma (TCGA, PanCancer Atlas), Lung Adenocarcinoma (TCGA, Provisional), Lung Adenocarcinoma (TSP, Nature 2008), and MSK-IMPACT Clinical Sequencing Cohort for Non-Small Cell. There were 16 samples (1%) with genetic alteration (Figure 1C) from 1530 profiled samples. The genetic alterations included 12 amplification, 3 missense mutations and one truncating mutation (Figure 1D and 1E).

èWe modified the Figure 1 and added the description in Line 95-98 of the Result part 2.1.

We also explored genetic alterations in the IGFBP7 locus via cBioPortal [17,18]. IGFBP7 was altered in 16 samples (1%) from among 1530 profiled samples in 7 datasets (Figure 1C). The genetic alterations in IGFBP7 included 12 amplification, 3 missense mutations, and 1 truncating mutation (Figure 1D and 1E).”

èThe legend of Figure 1 was added.

”(C) The oncoprint revealed that 1% (16 altered/1530 profiled) of LUAD patients had alterations in IGFBP7 in the lung adenocarcinoma cohort from cBioPortal. (D) Three missense mutations and 1 truncating mutation in the IGFBP7 locus (E) Alteration frequency in different datasets.”

èWe also added the description in the Method part 4.1.

”Additionally, we used cBioPortal to explore gene alterations in the candidate gene locus, including copy number variation and mutations [17,18]. We queried lung adenocarcinoma samples in 7 different datasets [45-48].”

4.    Was IGFBP7 expression correlated with other IGFBPs? IGFBP5 is listed as a highly ranked target in Figure 1B.  

R: To examine the expression correlation between IGFBP7 and IGFBP5, we obtained the expression datasets of lung cancer cell lines from Cancer Cell Line Encyclopedia (CCLE) and lung adenocarcinoma tissues from TCGA-LUAD. As illustration in following figure, there is no expression correlation between IGFBP7 and IGFBP5 in cell line dataset, but IGFBP5 expression is slightly correlated to IGFBP7 in tissue dataset.

è We added the description in Line 100-105 of the result part 2.1.

Additionally, IGFBP5 was listed as a highly raked candidate target gene. It was unknown whether a correlation exists in gene expression between IGFBP7 and IGFBP5. We obtained the gene expression datasets for lung cancer cell lines from the Cancer Cell Line Encyclopedia and lung adenocarcinoma tissues from The Cancer Genome Atlas-Lung Adenocarcinoma Disease Type. There was no expression correlation (r = -0.16) between IGFBP7 and IGFBP5 in the cell line dataset, while IGFBP5 expression was slightly correlated with IGFBP7 in the tissue dataset (r = 0.26) (Supplementary Figure S2).”

è We added Supplemental Figure S2.

Reviewer 4 Report

Through the data-set analysis and in vitro evaluation, the authors found that high IGFBP7 expression may drive resistance to EGFR-TKIs in lung cancers with EGFR mutations. Major and minor comments by the reviewer is as follows.

Lines 57-58, "Mutations in KRAS...." The reviewer believe that this is not a story for lung cancers with EGFR mutations.

Lines 76-78, please refer the all original works.

Line 100. The readers may not understand what PC/gef, HCC827/gef, and HCC4006ER cells are. The authors have to describe a brief details for these cells.

In Figure 2, the expression of HRASLS and RNF182 also looks good as potential mechanism for resistance. How was the results of these two molecules in the analysis of Figure 1?

For Figure 3, please add data for HCC827/gef, too.

Author Response

1.        Extensive editing of English language and style required 

R: We have done thorough English editing by Editage, a brand of Cactus Communications, and corrected the grammatical mistakes in the revised manuscript.

2.        Lines 57-58, "Mutations in KRAS...." The reviewer believe that this is not a story for lung cancers with EGFR mutations.

R: We deleted the sentence.

3.        Lines 76-78, please refer the all original works.

R: The original articles of the 4 GEO datasets were cited and they included

GSE80344(Oncotarget 2017 Nov 28;8(61):103340-103363.), GSE106765 (Sci Rep 2018 Oct 5;8(1):14896.) and GSE95558 (Mol Cancer Ther 2017 Aug;16(8):1645-1657.) The dataset of GSE103350 had no reference published paper.

è We inserted these original works in in-text citations in Line 77 and reference list [13-15].

4.        Line 100. The readers may not understand what PC9/gef, HCC827/gef, and HCC4006ER cells are. The authors have to describe a brief details for these cells.

R: We added a brief description for the three EGFR TKI-resistance cells.

èWe added the description in Line 116-118.:

“PC9/gef, HCC827/gef, and HCC4006/ER cells were selected from TKI-sensitive parental cells (PC9, HCC827, and HCC4006) that had been continuously exposed to increasing concentrations of the EGFR TKIs gefitinib or erlotinib.”

5.        In Figure 2, the expression of HRASLS and RNF182 also looks good as potential mechanism for resistance. How was the results of these two molecules in the analysis of Figure 1?

R: RNF182 is ranked as 57, but HRASLS is ranked as more than 10000.

6.        For Figure 3, please add data for HCC827/gef, too.

R: We added the data of HCC827/gef in the result. Analysis of cell viability and expressions of apoptotic proteins after gefitinib treatment in HCC827/gef cells were shown in Supplemental Figure S4.

èIn Line 157-165, We modified the sentence: “To examine the role of IGFBP7 in the resistance to EGFR-TKIs, IGFBP7-specific siRNA (si-IGFBP7-1 and si-IGFBP7-4) against IGFBP7 was used to knock down IGFBP7 in EGFR-TKI-resistant cells (PC9/gef, HCC4006/ER, and HCC827/gef). We found that transfection with IGFBP7-specific siRNAs suppressed IGFBP7 expression and recovered EGFR-TKI sensitivity in EGFR-TKI-resistant cells (PC9/gef, HCC4006/ER, and HCC827/gef; Figures 3A, 3B and Supplementary Figure S4A and S4B). The percentage of apoptotic cells, which was quantified as Annexin-V-positive cells, was significantly increased in IGFBP7-suppressed cells compared to in control cells (PC9/gef-si-scramble and HCC4006/ER-si-scramble) following exposure to gefitinib and afatinib (Figure 3C and Supplementary Figures S3A).”

èIn Line 168-172 of Page 6, We added:IGFBP7-knockdown-HCC2006/ER (HCC4006/ER-si-IGFBP7-1 and HCC4006/ER-si-IGFBP7-4) and IGFBP7-knockdown-HCC827/gef cells (HCC827/ER-si-iGFBP7-1) also showed increased levels of apoptosis markers compared to in HCC2006/ER-si-scramble and HCC827/gef-si-scramble cells, respectively (Supplementary Figures S3b and S4C).”

è We added Supplemental Figure S4:”Supplementary Figure S4 Knockdown of IGFBP7 expression reversed EGFR-TKI resistance in HCC827/gef cells by enhancing EGFR-TKI-induced cleave-PARP expression (A) HCC827/gef cells were transfected with IGFBP7 small interfering RNAs (siRNA; si-IGFBP7-1) or scramble siRNA (si-scramble). The effect of siRNAs was evaluated by quantitative RT-PCR (left) and western blot analysis (right). (B) Cellular viability of si-scramble and si-IGFBP7 transfectants was determined at different doses of gefitinib for 96 h using MTS assays. Error bars show the standard deviations for n = 3 independent experiments. (*** p < 0.001). (C) HCC827/gef was exposed to 50 and 250 nM of gefitinib for 24 h. Next, the apoptosis marker cleaved-PARP was assayed by western blot analysis. ” 

Round 2

Reviewer 4 Report

The authors modified adequately. However, some revised part say "HCC2006 cells" which might be a typographical error of "HCC4006 cells".